# Plant Dehydrins: Expression, Regulatory Networks, and Protective Roles in Plants Challenged by Abiotic Stress

**DOI:** 10.3390/ijms222312619

**Published:** 2021-11-23

**Authors:** Zhenping Sun, Shiyuan Li, Wenyu Chen, Jieqiong Zhang, Lixiao Zhang, Wei Sun, Zenglan Wang

**Affiliations:** Shandong Provincial Key Laboratory of Plant Stress Research, College of Life Science, Shandong Normal University, Jinan 250014, China; zhenpingsun0625@163.com (Z.S.); echosemail@163.com (S.L.); yuuuu_ccc1996@163.com (W.C.); zjq210306@163.com (J.Z.); LixiaoZhang0125@163.com (L.Z.)

**Keywords:** abiotic stress response, abscisic acid, calcium ion, dehydrin, mitogen-activated protein kinase

## Abstract

Dehydrins, also known as Group II late embryogenesis abundant (LEA) proteins, are classic intrinsically disordered proteins, which have high hydrophilicity. A wide range of hostile environmental conditions including low temperature, drought, and high salinity stimulate dehydrin expression. Numerous studies have furnished evidence for the protective role played by dehydrins in plants exposed to abiotic stress. Furthermore, dehydrins play important roles in seed maturation and plant stress tolerance. Hence, dehydrins might also protect plasma membranes and proteins and stabilize DNA conformations. In the present review, we discuss the regulatory networks of dehydrin gene expression including the abscisic acid (ABA), mitogen-activated protein (MAP) kinase cascade, and Ca^2+^ signaling pathways. Crosstalk among these molecules and pathways may form a complex, diverse regulatory network, which may be implicated in regulating the same dehydrin.

## 1. Introduction

Hostile environmental conditions such as high salinity, drought, and low temperature threaten plant growth and development and lower crop yield. Faced with stress, immobile plants have evolved complex regulatory mechanisms to adapt to or resist stress over a long period of natural selection. Thus, the self-protection mechanisms of plants have been investigated to improve plant abiotic stress resistance [1].

Abiotic stress can cause the accumulation of reactive oxygen species (ROS) which damage the structure of cell membranes and affect the functions of proteins and nucleic acids. In order to alleviate reactive oxygen species damage, plants have evolved natural antioxidant systems, enzymatic and non-enzymatic antioxidants. Diversiform antioxidant enzyme of plants, such as super-oxide dismutase (SOD) and ascorbate peroxidase (APX), have been reported to get rid of the ROS in abiotic stress response [2]. Non-enzymatic antioxidants mainly are organic compounds such as active peptides, vitamin E, vitamin C, polyphenols and anthocyanins. For example, rubiscolin-6, an opioid peptide of Rubisco that has the function of fixing CO_2_ in plant photosynthesis [3,4], serves as a radical scavenger [5]. Under drought treatment, the underground tissues of soybean express higher concentrations of vitamin E which protects the membrane from peroxidation [6]. Here, we will introduce a class of LEA protein family members-dehydrins, which also exhibit anti-oxidant functions.

Late embryogenesis abundant (LEA) proteins were initially detected in the late stages of seed maturation and were later found in seedlings, roots, stems, and other plant organs as well. Studies have shown that LEAs are implicated in plant growth, development, and stress response [7,8]. Dehydrins are Group II LEA proteins that play vital roles in establishing and maintaining plant stress tolerance. Several dehydrins, including RD29A, RD29B, and RAB18, show dynamic changes in their expression in response to plant abiotic stress; these changes are indicators of plant abiotic stress tolerance. The recent discovery of interactions between dehydrins and other proteins demonstrated the diversity of their physiological functions [9,10,11].

There are many excellent summaries on the roles of dehydrins. Yang et al. have summarized the different molecular mechanisms of dehydrins response to environmental stress in plants [12]. Poonam et al. provided a linked and comprehensive introduction of dehydrins gene family. Over the past 30 years, dehydrins have been known as chaperones which could protect plant from cold and free radical. Recent years we known dehydrins participate in regulation of genes that response to abiotic stress. Interestingly, dehydrins can also participate in histone modification [13]. However, as we know, few reviews have elaborated on dehydrin gene regulation at upstream signaling pathway. Dehydrin gene expression has been traditionally categorized as ABA-dependent and ABA-independent. Nevertheless, the existence of a crossing point between two pathways, some subsequent studies found. Therefore, dehydrin expression may, in fact, be regulated by multiple signal transduction pathways. The present review focuses on the complex dehydrins gene expression regulatory network and aims to elucidate the self-protection mechanism that dehydrins activate in plants challenged by abiotic stress.

## 2. Dehydrin Structure, Classification, and Localization

Dehydrins (DHNs) are thermostable, highly hydrophilic proteins with molecular weights in the range of 9–2000 kDa. They contain a characteristic lysine-rich K-segment near the *C*-terminus that can form an amphiphilic α-helix conferring them with high hydrophilicity. A previous study showed that the K-segment binds the dehydrins to cell membranes [14] and protects LDH and antioxidont enzymes [15,16]. There are also other conserved motifs in the DHNs, containing the Y-segment and S-segment. Based on the presence of the K-, S-, and Y-segments, DHNs are classified into the Kn, SKn, KnS, YnKn, and YnSKn structural subgroups. Dehydrins have been detected in various plants. YSK_2_-type RAB17 was found in maize, SK_3_-type COR410 and YSK_2_-type DHN5 occurred in wheat, YSK_2_ type ECP40 was identified in carrot, and Y_2_K_4_-type CAS31 was observed in *Medicago truncatula*. Researchers have identified four dehydrin family genes from cucumbers, namely YnKn-type CsDHN1, YnSKn-type CsDHN2 and CsDHN4, and other SKn-type CsDHN3, all of which exhibit high proportion of alpha helix and random coils, similar to other plant dehydrins, and the other three dehydrins are all induced by drought, salt and ABA [17]. There are numerous dehydrins in the common model dicotyledon Arabidopsis and they are widely applied in stress biology. They include RAB18 (Y_2_SK_2_-type), RD29A (Kn-type), RD29B (Kn-type), COR47 (SK_3_-type), ERD10 (SK_3_-type), ERD14 (SK_2_-type), and Lit30 (K_6_-type). *RD29A*, *RD29B*, *RAB18*, *COR47*, and others are both dehydrin and abiotic stress marker genes.

Dehydrins abound in mature seeds and enable them to tolerate drought. In situ hybridization (ISH) revealed that *MdoDHN11* was expressed in the central nucellus parenchyma layer between the endosperm and the testa in apple seeds [18]. Dehydrin localization changes with embryonic maturation. Maize *RAB17* mRNA initially appeared in the embryo axis, was subsequently detected in the scutellum, and finally accumulated in the axis cells and provascular tissues [19]. Dehydrins may be found in many subcellular locations including the nucleus and the cytoplasm. Confocal microscopy disclosed that AtCOR47 and AtERD10 were localized to the cytoplasm whereas AtRAB18 was distributed in both the nucleus and the cytosol [10]. Maize RAB17 protein is distributed in the cytoplasm and nucleus, meanwhile its nuclear localization depends on nuclear localization signal (NLS) peptide binding via autophosphorylation [19]. Nuclear dehydrin localization was also reported for the embryonic axes of developing beech seeds [20]. Nuclear dehydrins are also regarded as protective substances in scattered chromatin [20]. Dehydrins may also occur in the plasma membrane and certain organelles such as the mitochondria and chloroplasts. AmDHN132 and AmDHN200 proteins were detected in the plasma membrane of transgenic Arabidopsis. Hence, they play roles in membrane protection [21].

## 3. Dehydrins Participate in Plant Response to Abiotic Stress

Dehydrins are Group II LEA proteins and play crucial roles in plant adaptation to abiotic stressors such as drought, cold, and salinity. There is substantial evidence that abiotic stress induces dehydrin expression.

### 3.1. Dehydrin Expression under Drought Condition

In many regions, plants are prone to drought during the summer season. Drought causes dehydration and cell damage and dehydrin genes are induced by dehydration stress. In white spruce, eight dehydrins responded to drought stress. The transcription levels of *PgDHN10*, *PgDHN16*, *PgDHN33*, and *PgDHN35* increased several fold after a few days of water shortage [22]. Lv identified the novel YSK2-type dehydrin CdDHN4 in the Tifway and C299 bermudagrass varieties. *CdDHN4* in the root and shoot was strongly upregulated by drought [23]. The dehydrin accumulation patterns varied with plant developmental stage during drought stress. The dehydrin profiles of winter wheat at the seedling, tillering, jointing, and anthesis stages differed from each other [24].

### 3.2. Dehydrin Accumulation during Cold Stress

Low temperatures may cause chilling or freezing injury in plants. Low temperature upsets the balance between root water absorption and leaf water transpiration, thereby causing dehydration. At subzero temperatures, extracellular ice crystals cause cellular dehydration and ultimate freezing injury. Plants challenged by water deficits and cold stress express cytoplasmic dehydrins in various tissues. Vítámvás et al. studied dehydrin accumulation in wheat and barley fields during wintertime. Relative dehydrin accumulation in winter wheat WCS120 and barley DHN5 was positively correlated with wintertime crop survival. Dehydrins and their transcripts might serve as cold tolerance markers [25]. In *Arabidopsis thaliana*, *ERD10*, *ERD14*, and *Lti30* accumulate at low temperatures [26,27]. Research demonstrated that a combination of dehydrins and membranes improves plant cold tolerance. The K-segment of Lti30 combined with lipid head groups [14] via electrostatic interactions. In this manner, lipid and bound protein molecule mobility was limited and aggregates formed to protect the cell membrane [28].

### 3.3. Dehydrin Expression Is Induced by Salt Stress

High salt concentrations cause osmotic change, inhibit root water absorption, and eventually damage plant cells. Plants protect their cells from excessive water loss by accumulating dehydrins. As these proteins are highly hydrophilic, they bind large amounts of water and reduce water loss. High salinity induces dehydrin accumulation in the roots, stems, leaves, and seeds. Hernán et al. detected at least four dehydrins (30 kDa, 34 kDa, 50 kDa, and 55 kDa) in *Chenopodium quinoa* embryos subjected to salt stress [29]. Salt treatment altered *CaDHN* expression. Relative to the untreated control, the salt-stressed plants presented with 134-fold and 420-fold increases in *CaDHN5* and *CaDHN7* expression and >20-fold increases in *CaDHN1*, *CaDHN2*, and *CaDHN3* expression [30]. A reversed genetics approach was used to explore the impact of dehydrin genes on plant salt tolerance enhancement. Under high NaCl concentrations, transgenic *Arabidopsis thaliana* overexpressing pepper *CaDHN4* and *CaDHN5* exhibited higher seed germination rates than WT plants [31,32]. *GhDHN_03* and *GhDHN_04* knockdown demonstrated the putative roles of dehydrins in augmenting osmotolerance and salt tolerance in cotton [33].

## 4. Dehydrin Gene Expression Regulatory Networks

Dehydrin expression is affected by various environmental factors. Drought, low temperature, and salinity rapidly induce dehydrin accumulation. Under abiotic stress, the regulation of dehydrin gene expression involves several classic signal transduction pathways, including ABA, MAPK, and Ca^2+^ (Figure 1), all of which activate transcription factor (TF) binding to the cis-acting elements in dehydrin promoter [34], thereby exerting various regulatory functions.

### 4.1. ABA Is a Dehydrin Mediator in Stress Response

ABA is also called “anti-adversity hormone”, and exogenous ABA treatment induces the expression of various dehydrins. Hence, the latter are considered responses to ABA proteins (RAB). Mundy and Chua identified the dehydrin gene *RAB21* in rice roots, leaves, embryos, and cell suspensions treated with ABA (10 μM) or NaCl (200 mM) [35]. *RAB18* mRNA accumulated in *Arabidopsis thaliana* exposed to exogenous ABA, cold, or water stress [36]. Subjecting wild type *A. thaliana*, an ABA-insensitive *A. thaliana* mutant (*abi1*), and an ABA-deficient *A. thaliana* mutant (*aba-1*) to low temperature demonstrated that the mutants were defective in the induction of *RAB18*. In the *abi1* mutants, *RAB18* expression was delayed. In the *aba-1* mutants, it was absent altogether [37]. Several cold stress-responsive dehydrin genes in *A. thaliana* such as *Lti30* [38], *COR39*, and *COR47* [39] may respond to exogenous ABA. Abscisic acid has various effects on the induction of different dehydrins. YnSKm-type *DHN* (YSK2-type *TaDHN17*) upregulation was detected in the leaves and roots of ABA-treated wheat seedlings whereas KS-type *DHN* (*TaDHN1*) expression was constant regardless of ABA treatment [40]. Therefore, dehydrin gene expression is regulated by both ABA-dependent and ABA-independent signaling pathways. Both types of pathways involve Ca^2+^ signaling which induces dehydrin gene expression under salinity, drought, and low temperature stress. Next, we will discuss the important roles of ABA core signal transduction pathways in dehydrin gene expression regulation.

The ABA signaling pathway is critical in plant drought and salt stress response. ABA receptors have been identified and core ABA signaling pathways have been elucidated [41]. Various models have been proposed [41,42,43], including ‘gate’ and ‘latch’ model (Figure 2). These findings may facilitate the study of dehydrin gene expression regulation. PYLs (PYR/PYL/RCAR) and PP2Cs are ABA co-receptors [44]. ABA binds PYLs to form ABA-PYL complexes which, in turn, bind PP2Cs, inhibit their protein phosphatase activity, and release the SnRK2s that the PP2Cs had inhibited [45,46]. SnRK2s are activated by autophosphorylation. Thence, they phosphorylate numerous downstream ABA response effector proteins [47]. ABI5 (ABA-insensitive 5) and AREBs (ABA-responsive element binding proteins)/ABFs (ABA-responsive element-binding factors) in bZIP TFs are phosphorylated SnRK2 substrates [48,49]. A yeast two-hybrid (Y2H) system verified the interactions between FtbZIP83 and FtSnRK2.6/2.3 [50]. Bimolecular fluorescence complementary (BiFC) analysis showed that AREB1, AREB2, and ABF3 interact with SnRK2s [51]. ABA-responsive elements (ABREs) were detected in the dehydrin gene promoter regions of several different plant species. The OsDHNN-RAB16D promoter comprises eight ABA-responsive cis-acting elements [52]. WZY2 from wheat interacts with PP2C and its promoter contains several putative stress-related ABREs [53,54]. Thus, *DHN* expression may be regulated by ABA core pathways. *CsSnRK2.5* overexpression conferred hypersensitivity to exogenous ABA. Transgenic Arabidopsis presented with elevated dehydrin levels compared to WT plants. Therefore, CsSnRK2.5 is a positive regulator of ABA-mediated dehydrin expression [55]. *VvABF2* is a homolog of AREB/ABFs from Arabidopsis that was isolated from grapevine and constitutively expressed in Arabidopsis. This discovery confirmed the vital role of the AREB/ABF-SnRK2 pathway. Compared with WT Arabidopsis plants, *VvABF2* transgenic plants were more sensitive to exogenous ABA and exhibited higher *RAB18*, *LEA*, and *RD29B* expression levels and osmotolerance [56]. *IbABF4* overexpression clearly improves expression level of *RD29A, RD29B, RAB18*, and *RD22* [57]. However, the *areb1*-*areb2*-*abf3* triple mutant was relatively more resistant to ABA. Its stress-responsive genes (*RAB18* and *RD29B*) were downregulated and it displayed poor drought tolerance [51]. These findings suggested that the AREB/ABF-SnRK2 pathway plays a vital role in the dehydrin-regulated pathway under drought and salinity stress conditions. MYB TFs also regulate ABA on dehydrins. AtMYB44 interacts with the ABA receptor RCAR/PYL9, competes for the ABA receptor and ABI1 binding sites, attenuates the inhibitory effect of ABI1, and downregulates *RAB18* [58]. Cotton *GhMYB73* was rapidly induced by ABA and salinity and the transcription levels of *AtPYL8*, *AtABF3*, and *AtRD29B* were significantly increased in its overexpressing lines. GhMYB73 interacted with AtPYL8 and its homolog GhPYL8. Hence, GhMYB73 may regulate ABA on dehydrins via PYL8 [59].

SnRK2s also regulate the expression of certain ABA-independent *COR* (cold-regulated) genes under cold stress [60]. OST1 (SnRK2.6) kinase is activated by low temperature stress and phosphorylates ICE1 which inhibits HOS1-mediated ICE1 ubiquitination and degradation. OST1 also competes with HOS1 for ICE1 binding and releases ICE1 from the HOS1-ICE1 complex. OST1 stabilizes ICE1 and the latter gradually accumulates and recognizes the C-repetitive binding factor (CBF) promoters [60,61]. The latter are DNA regulatory elements in the *COR* promoter and contribute to plant cold adaptation [62]. CBF1, CBF2, and CBF3 are cold-induced CBF TFs in the Arabidopsis genome that bind CRT/DRE/LTRE [63]. Various dehydrins have been detected in cold-regulated proteins such as COR15a whose promoters contain CRT/DRE/LTRE elements [34]. Compared with the WT, *CBF* and downstream cold response genes such as *AtRD29A*, *AtCOR15A*, and *AtCOR47* in the *DlICE1-* and *SmICE1a* overexpression lines were upregulated at low temperature and enhanced plant cold tolerance [64,65]. It was proposed that *PaICE1* activates *PaCBF1* by binding MYC motifs in the *PaCBF1* promoter, upregulating the downstream candidate gene *PaDHN1*, and protecting plants from cold damage [66]. CBFs are regulated by ICE1 and are affected by phytochrome-interacting factor under cold stress. A recent model showed that CBF proteins accumulated under cold stress stabilized phytochrome B by interacting with phytochrome-interacting factor3 (PIF3), and cold-stabilized phyB mediated PIF1, PIF4, and PIF5 degradation, thereby improving cold-regulated gene expression. Thus, *COR* expression may be coordinated by multiple TFs [67].

Regulation of gene expression by the DRE/CRT cis-element might be ABA-independent whereas gene expression regulated by the ABRE element may be ABA-dependent [68]. However, a recent study showed that there is cross-talk between these pathways in terms of gene expression regulation. The *COR47* promoter contains two CRT/DRE/LTREs and one ABRE and its expression is induced by both cold and exogenous ABA [27]. ABA and LT synergistically upregulate the VvCBF2, VvCBF3, VvCBF4, and VvCBF6 TFs and promote antioxidant and dehydration gene expression and grape bud cold adaptation [69]. Certain DRE/CRT motifs also participate in the ABA-dependent pathway. In Arabidopsis, drought and ABA induce CBF4 expression. Constitutive *CBF4* overexpression upregulates drought- and cold-related downstream genes containing DRE/CRT cis-elements such as *COR47* and *COR78*/*RD29A* [70]. Under abiotic stress, *OsDREB1F* overexpression upregulated *RD29A* and *COR15a* with promoters bearing the DRE sequence and induced *RD29B* and *RAB18* with promoters containing the ABRE (but not the DRE) element [71].

### 4.2. The Calcium Signaling Pathway Has Multiple Regulatory Effects on Dehydrin Expression

Abiotic stress conditions such as high salt, drought, and low temperature alter plant cell Ca^2+^ concentrations. Ca^2+^ is a second messenger that transmits external signals to plant cells and transduces the signals downstream to regulate various cellular processes such as gene expression via Ca^2+^ sensors. The latter consist mainly of CDPKs (calcium-dependent protein kinases), CBLs (calcineurin B-like proteins), CMLs (calmodulin-like proteins), and CaMs (calmodulins) which have complex EF hand domains that bind Ca^2+^. The calcium signal pathway has a wide range of regulatory effects on dehydrin expression (Figure 3).

CDPKs comprise serine/threonine protein kinase, autoinhibitory, and CaM-like domains [72,73]. Increases in Ca^2+^ concentration promote Ca^2+^ binding to the *N*-terminus of the CML domain, change the conformation and shift of the autoinhibitory domain, and lead to intramolecular autophosphorylation and CDPK kinase activation [72]. Researchers identified 34, 26, and 27 members of the *CDPK* family in Arabidopsis, tea, and cassava [74,75,76], respectively. A previous review reported that CDPKs phosphorylate TFs in response to multiple abiotic stress-related pathways [77]. A current study disclosed that wheat TaCDPK9-1 (*Triticum aestivum* L.) activates TabZIP 8, TabZIP 9, and TabZIP 13 and regulates ABA synthesis and accumulation under salt stress. Thus, CDPKs might affect dehydrin gene expression downstream of ABA [78]. Arabidopsis CPK4, CPK11, and CPK32 kinases positively regulate Ca^2+^ implicated in ABA signaling by phosphorylating ABA-responsive bZIP TFs such as ABF1 and ABF4 [79,80]. These transcription factors bind the cis-acting regulatory elements of target genes and promote dehydrin response to abiotic stress. ABA strongly induced *ABFs* and *RAB18* in transgenic *CPK4* and *CPK11* overexpression lines [80]. SnRKs also phosphorylate ABF TFs. Therefore, multiple kinases may regulate downstream target gene expression by acting on common substrates in the ABA signaling pathway. RT-PCR analysis revealed that *AtCPK1* knockdown and overexpression inhibits and promotes *RD29A* and *COR15A* expression, respectively, under salinity and drought stress [81]. The foregoing results indicate that *CDPK* family members positively regulate dehydrin expression and help plants contend with abiotic stress.

CBLs combine with their downstream gene family CIPKs to decode the calcium signal and initiate the stress response mechanism [82]. Seven and five *CBLs* and 20 and 15 *CIPKs* were identified in wheat and eggplant, respectively, and the interactions between these protein types were confirmed by Y2H and BiFC assays [83,84]. Differential *CBL* expression under abiotic stress shows that this gene in implicated in the responses to various stressors such as drought, salinity, low temperature, and ABA. Recent research demonstrated that *CBLs* were differentially expressed in transgenic rice seedlings subjected to drought and salt stress [85]. An overexpression analysis verified that *OsCBL3* and *OsCBL8* positively regulated salt tolerance while *OsCBL5*, *OsCBL6*, and *OsCBL7* improved drought tolerance in rice. The Arabidopsis salt overly sensitive (SOS) pathway SOS3-SOS2 is a typical CBL-CIPK module also known as CBL4-CIPK24. It maintains intracellular Na^2+^ balance under high salt stress [86,87]. Numerous TFs regulating dehydrin gene expression have been identified as putative downstream signal components of the CBL-CIPK module. Under abiotic stress, *CBL* and *CIPK* overexpression and knockdown alter stress marker gene expression patterns. In Arabidopsis ectopically overexpressing *TaCIPK27* and subjected to drought stress, *RD29B* and *SnRK2s* related to ABA signaling and *DREB* genes involved in drought stress response were significantly upregulated and improved drought tolerance [88]. The *COR47* and *KIN1* expression levels were markedly higher in *TaCIPK23* overexpression lines than they were in the WT [89]. In *CIPK3* T-DNA insertion mutants, cold-induced *RD29A/COR78* expression was delayed and ABA-responsive *RD22* and *RD29B* were downregulated [90]. The calcium-mediated CIPK3 pathway is a cross-node linking cold stress and the ABA signal transduction pathway. Li et al. discovered Arabidopsis Yin Yang 1(*AtYY1*) which is a new negative regulator of ABA response that is positively regulated by *ABA* INSENSITIVE4 (*ABI4*). It binds the *ABA REPRESSOR1 (ABR1)* promoter, upregulates *ABR1*, and downregulates *RD29A*, *RD29B*, and *COR15A* [91]. Other researchers demonstrated that the CBL9-CIPK3 module acts upstream of ABR1 and may fine-tune ABA-responsive gene expression [92,93].

CML proteins also have an EF hand structure which is a common calcium ion receptor in plants. Nevertheless, their biochemical characteristics are highly diverse among plant species [94]. Four *CaMs* and 36 *CMLs* were identified in a whole-genome analysis of woodland strawberry (*Fragaria vesca*). The simultaneous conservation and divergence of the *FvCaM* and *FvCML* gene structures resulted in functional similarities and differences [95]. Eighty-three *CMLs* were detected in the apple genome and were widely distributed in various tissues and organs. Most of them were induced by phytohormones and abiotic stress [96]. Of the 50 *CMLs* in Arabidopsis, CML20 negatively regulates dehydrin expression and ABA-induced stomatal movement. Drought and ABA upregulated stress response genes such as RAB18, ERD10, COR47, and RD29A in a cml20 mutant [97].

CaMs lack kinase activity. They must be combined with Ca^2+^ and form Ca^2+^-CaM complexes to alter molecular conformations and activate various downstream target proteins. The seven *CaMs* were found in the Arabidopsis genome [98], and four *MdCaMs* were recently identified in apple [99]. The calmodulin-binding transcriptional activator (CAMTA) family includes various TFs in the bZIP and MYB families. They contain DNA-binding domains and interact with CaMs to regulate the expression of downstream genes affected by stress signals and ABA [100,101]. However, CAMs may also regulate cold-induced gene expression. By contrast, CAMTA3 promotes cold-induced gene expression by upregulating CBF2 [102]. Neither CAMTA3 nor CAMTA5 is affected by gradual temperature decreases. However, precipitous declines in temperature cause them to promote *CBF1* expression [103]. When Arabidopsis is subjected to cold, Ca^2+^/CaM interacts with AtCRLK1, activates the MAPK signal pathway, and upregulates *COR* [104,105]. Nevertheless, CaMs may also indirectly function as negative *COR* expression regulators. *CaM3* overexpression in Arabidopsis repressed both *RD29A* and *COR6.6* [106].

### 4.3. The MAPK Cascade Pathway Acts Upstream to Regulate Dehydrin Gene Expression

Mitogen-activated protein kinase (MAPK) family members are involved in many biological processes and are crucial regulators of plant hormones and stress responses [107]. The MAPK cascade pathway includes the kinases MAP3K (MAPKKK or MEKK), MAP2K (MKK or MEK), and MAPK which are activated by sequential phosphorylation. Under cold stress, Ca^2+^ combines with CaM, activates the MAP cascade pathway, and upregulates cold-responsive genes [104,105,108]. In this process, MPK3 and MPK6 phosphorylate ICE1, thereby destabilizing it and decreasing its transcriptional activity [109]. ICE1 is subsequently ubiquitinated by E3 ligase and degraded in the proteasome. Thence, the CBF-controlled *COR* is downregulated [104]. In *mpk3* and *mpk6* single- and double mutants, *CBFs*, *COR15A*, and *RD29A* trend higher levels, suggesting the positive effect of MAPK3/6 on dehydrin genes [108]. However, calcium/calmodulin-regulated receptor-like kinase 1 (CRLK1) is upregulated by calcium-calmodulin and activates the MEKK1-MKK2-MPK4 pathway [104,105]. MPK4 inhibits MPK3/6 activity [109]. Hence, the MEKK1-MKK2-MPK4 module plays an active upstream role in adjusting *COR* expression and cold tolerance (Figure 4).

The ABA core pathway can activate MAPKs. PYR/PYL/RCAR-SnRK2-PP2C, known as the core ABA component, stimulates the MAP3K17/18-MKK3-MPK1/2/7/14 cascade pathway while the plant is subjected to drought and ABA stress [110]. MAP3K phosphorylates SnRK2.6/OST1 which are essential for ABA-triggered SnRK2 activation and promote TF phosphorylation and SLAC1 activation [111]. A transcriptome analysis revealed that MAP3K18 levels continued to rise throughout salt stress [112]. Mitula et al. demonstrated direct interactions among MAP3K18, PP2C phosphatase ABI1, and activation by ABA. This mechanism controls the expression of the ABA-regulated dehydrin genes *RD29B* and *RAB18* [113]. This discovery corroborates the hypothesis that the MAPK cascade acts upstream of dehydrin expression (Figure 4).

In the dehydrin expression regulatory mechanism challenged by abiotic stress, various signaling pathways decode the external stress signals, transmit them downstream, regulate the expression of stress response and dehydrin genes, and mediate plant stress adaptation. Each signaling pathway may act on different targets, crosstalk occurs among various signaling pathways, and both phenomena contribute to the formation of a complex and diverse regulatory network.

## 5. Functional Diversity of Dehydrins

### 5.1. Dehydrins Protect Seeds from Dehydration during Maturation

The plant water content often decreases during seed ripening. This mechanism is determined by plant genetics and is accompanied by *LEA* expression [114,115]. Dehydrins comprise a LEA subfamily and are usually synthesized and accumulated in the late stages of seed maturation. In this manner, they confer dehydration resistance to the seeds. The 26-kDa and 44-kDa dehydrins in *Fagus sylvatica* seeds determine the hypocotyl drying level during seed development [116]. In apple seed formation, DHN11 accumulates in the maternal nucleus and protects the embryo and endosperm against desiccation [18]. Dehydrins may stabilize cell membranes and proteins, thereby ensuring seed survival during maturation and drying.

### 5.2. Dehydrins Stabilize Plasma Membranes

Dehydrins are intrinsically disordered proteins that are induced by dehydration stress and low temperature. The structure and phase properties of the lipid bilayer of the plasma membrane are closely linked to temperature and water content. Extensive research has been conducted on the effects of dehydrins on the plasma membrane. The typical K-segments of dehydrins display high membrane affinity. During cold stress, the K-segment of Lti30 had a high membrane vesicle affinity. Binding was regulated by pH-dependent His and phosphorylation, reduced the main lipid phase transition temperature, and stabilized the membrane [28,117]. An atomic-level analysis disclosed that the K-segments were partially folded into α-helical segments on the membrane surface [14]. Dehydrins also interact with plasma membrane intrinsic proteins (PIPs) but do not bind membrane lipids. Plant PIPs are important members of the aquaporin (AQP) family. They are located on the membrane where they control abiotic stress-related water transport [118,119]. Hernandez-Sanchez et al. reported a homodimer interaction between AtCOR47 and AtERD10 [10]. AtCOR47, AtERD10, and AtRAB18 formed a heterotrimeric complex with OpsDHN1 from *Opuntia streptacantha*. Their mutual interactions protect proteins against abiotic stress damage [10]. BiFC technology revealed that AtPIP2Bs (Arabidopsis plasma membrane PIP family aquaporins) are common targets of OpsDHN1, AtCOR47, AtERD10, and AtRAB18 [11]. The interactions between dehydrins and AtPIP2B may prevent AQP denaturation and inactivation and stabilize the plasma membrane and its protein components under abiotic stress. Recently, it was demonstrated that under drought stress, the dehydrin MtCAS31 promotes autophagous degradation of MtPIP2;7 which is a negative regulator of the drought response. In this manner, it decreases root water loss and improves drought tolerance [120]. Hence, dehydrins have different effects on the plasma membrane components. Nevertheless, the mechanisms by which they exert their various protective effects remain to be elucidated.

### 5.3. Dehydrins Protect Enzymatic Activity

Abiotic stress induces ROS generation and accumulation in plant cells, thereby causing secondary oxidative stress and nucleic acid, protein, and lipid damage. Plants have evolved numerous defense mechanisms to maintain cellular redox homeostasis. For instance, plant antioxidant enzymes scavenge ROS. Dehydrins protect enzyme activity against oxidative stress. Under osmotic stress, ERD14 activated both glutathione transferase Phi9 and catalase [16,121]. The K-segments of DHNs protected lactate dehydrogenase (LDH) activity [15,16]. The AtHIRD11 His residue protected LDH against inactivation by heavy metals [122]. The hydrophobic residues of the F-segments in Arabidopsis COR47 (FSKn-type dehydrin) protected LDH against inactivation by freezing [123]. Thus, dehydrins are multifunctional enzyme protectors and improve plant stress resistance [124]. Overexpression of dehydrin genes in many crops, such as wheat *DHN5*, rice *DHN1*, maize *DHN11*, and cucumber *LEA-S*, activates the plant antioxidant enzyme system, enhances the ability to remove active oxygen, and improves the stress tolerance ability [16,125,126].

### 5.4. Dehydrins Bind Metal Ions and DNA

When plants suffer the effects of environmental stressors, metal ions in metalloproteins are released and promote ROS production. Several plant dehydrins combine with metal ions, reduce DNA and protein damage caused by ROS, and alleviate physiological disorders caused by heavy metals. Y2SK2 and YSK3 dehydrins from *Agapanthus praecoxcan* bound Co^2+^, Ni^2+^, Cu^2+^, and Fe^3+^ [124]. *Vitis riparia* DHN1 had strong binding affinity for Zn^2+^, Ni^2+^, Cu^2+^, and Fe^2+^ but not for Mg^2+^ or Ca^2+^ [127]. Arabidopsis KS-type dehydrins reversed the conformational changes in LDH caused by Cu^2+^ ions and restored enzyme activity more effectively than either bovine serum albumin (BSA) or lysozyme [122]. As dehydrins are also localized to the nucleus, they also protect nucleic acids. The S- and NLS-segments of ZmDHN13 localized it to the nucleus [16]. Nuclear VrDHN1 binds DNA. Its lysine-rich K-segment is positively charged and may bind the negatively charged phosphate backbone of DNA, thereby protecting it against ROS [127]. Table 1 is an overview of the functions of various plant dehydrins in abiotic stress response.

## 6. Conclusions and Perspectives

The present review focused on the regulatory effects of ABA, MAPK, and second messenger calcium on dehydrin expression in plants under abiotic stress. This complex process involves diverse proteins and transcription factors. This review also addressed the physiological functions of dehydrins. However, several functional mechanisms of dehydrins remain to be elucidated. For example, hydrogen peroxide enters the plant cell via plasma membrane-intrinsic proteins (PIPs). It is unknown whether the interaction between dehydrins and PIPs affects H_2_O_2_ transport and participates in antioxidant protection.

Comprehensive analysis of the dehydrin gene expression regulatory network will increase our understanding of abiotic stress tolerance that developed during plant ecotype evolution. This finding will help clarify the plant self-protection mechanism and aid in the development and cultivation of high-yield crops that can tolerate adverse external environments.

## Figures and Tables

**Figure 1 ijms-22-12619-f001:**
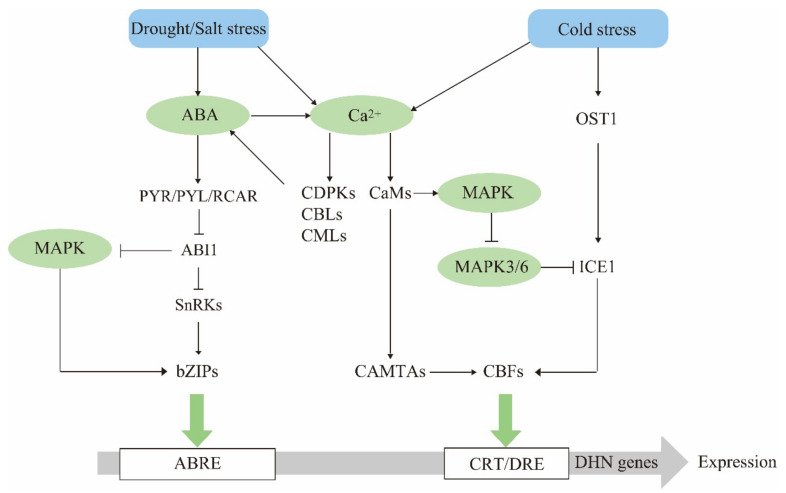
Dehydrin gene expression regulatory network model. Abiotic stressors such as drought, high salinity, and cold trigger dehydrin expression in plants depending on various signaling pathways. Drought and high salinity activate the ABA and calcium signaling pathways. After ABA binds its corresponding receptor, it acts on bZIP TFs and mediates dehydrin expression. Calcium ions are decoded by proteins such as CaMs, CDPKs, CBLs, and CMLs and regulate dehydrin expression via CBF TFs. They also act as bridges connecting drought/salinity and cold stress responses. ICE1 protein occurs in the early cold stress pathway. The CBFs in this pathway regulate the expression of multiple downstream CORs. The MAPK cascade also occurs throughout the entire regulatory network and regulates dehydrin expression upstream. Arrows and bars indicate positive and negative regulation, respectively.

**Figure 2 ijms-22-12619-f002:**
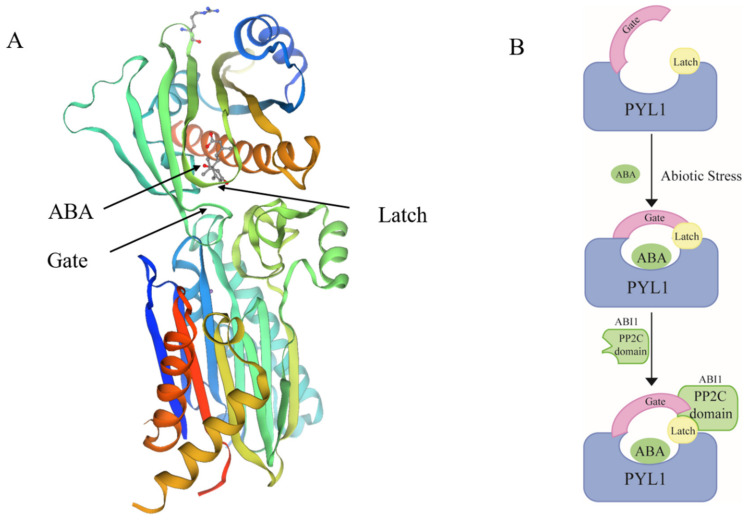
The interaction mechanism of ABA and ABA co-receptors. (**A**) The structure of the PYL1-ABA–ABI1 complex. The model and data come from the website (https://www.sib.swiss/, accessed on 17 November 2021). (**B**) The model of ‘gate’ and ‘latch’. At first, the ABA receptor PYL1 in an open state. When the abiotic stress signal comes, ABA enters the receptor pocket and the gate loop is closed. Then interact with the PP2C domain of ABI1 to inhibit their protein phosphatase activity, causing activation of the SnRK2s-dependent phosphorylation pathways.

**Figure 3 ijms-22-12619-f003:**
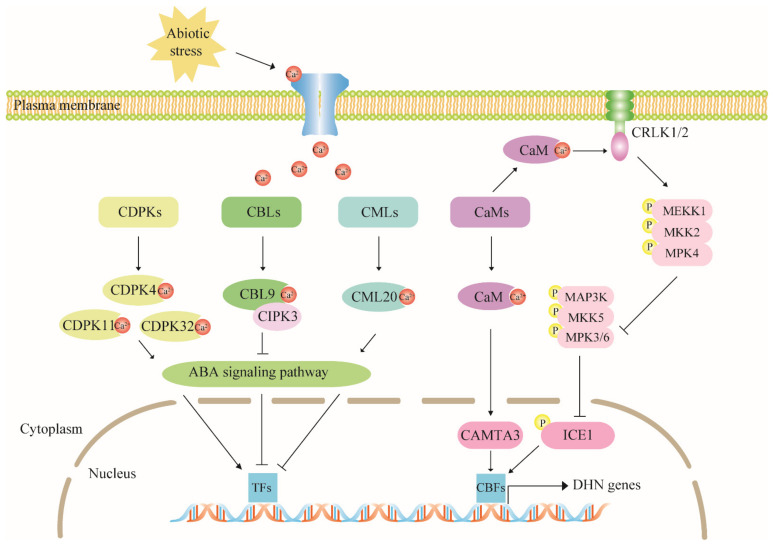
The calcium signal pathway has a wide range of regulatory effects on dehydrin expression. Sensors in the plasma membrane are stimulated by abiotic stress and promote the production of the secondary messenger Ca^2+^. Changes in calcium concentration are perceived by the calcium sensor proteins CDPKs, CBLs, CMLs, and CaMs. The latter either participate in the ABA signaling pathway or initiate phosphorylation/dephosphorylation cascade reactions of TFs such as CAMTAs and CBFs that eventually participate in dehydrin gene activation. CRLK1/2 regulates *COR* in the upstream of the MEKK1-MKK2-MPK4 cascade pathway. We use arrows and bars showing positive and negative adjustment, respectively.

**Figure 4 ijms-22-12619-f004:**
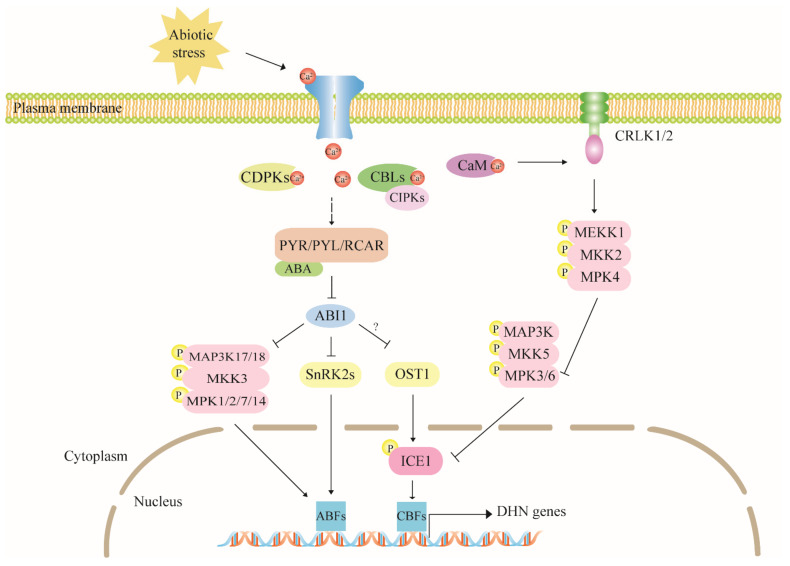
The MAPK cascade pathway is an upstream regulator of dehydrin gene expression. Under drought conditions, MAP3K-MKK3-MPK1/2/7/14 is activated by the core ABA component and regulates the expression of downstream dehydrins including RD29B and RAB18 by phosphorylating ABF TFs. In response to chilling, the MEKK1-MKK2-MPK4 cascade pathway is activated by Ca-CaM, counteracts the inhibitory effect of MAPK3/6 on ICE1, and upregulates downstream *CORs*. OST1 involved in the ABA signaling pathway can phosphorylate ICE1 to prevent its degradation and induce its accumulation. However, it is unclear whether ABA participates in this pathway. We use arrows and bars showing positive and negative adjustment, and solid and dashed lines indicate direct and indirect/unknown adjustment, respectively.

**Table 1 ijms-22-12619-t001:** Roles of dehydrins in abiotic stress tolerance in various plant species.

Species	Dehydrin	Abiotic Stress Responses	Functions	References
*Arabidopsis thaliana*	RAB18	Cold/drought/salinity	Freezing tolerance	[37]
	COR47	Cold	LDH cryoprotection	[123]
	Lti30	Cold	Membrane stabilization	[14,28,117]
	ERD10	Cold/drought/salinity	Oxidoreductase protection, membrane stabilization, chaperone activity	[121,128,129]
	ERD14	Cold/drought/salinity	Same as above	[121,128,129]
*Agapanthus praecox*	DHN (Y2SK2)	Cold/drought/salinity	Metal ion binding, enzyme activity protection, membrane cryoprotection	[124,130]
	DHN (SK3)	Cold/drought/salinity	Same as above	[124,130]
*Ammopiptanthus mongolicus*	DHN132	Cold/salinity	Membrane-protection	[21]
	DHN154	Cold/salinity	Same as above	[21]
	DHN200	Cold/salinity	Same as above	[21]
*Capsicum annuum*	DHN3	Drought/salinity	Upregulation of antioxidant enzyme system	[131]
	DHN4	Cold/salinity	Cell membrane stabilization, prevention of lipid peroxidation, inhibition of ROS accumulation	[32]
	DHN5	Salinity	Antioxidant capacity improvement	[31]
*Cerastium arcticum*	DHN	Cold/drought/salinity	Oxidative stress tolerance	[132]
*Cucumis melo*	*LEA-S*	drought/salinity	APX and CAT activity enhancement	[133]
*Cucumis sativus*	DHN4	Cold/ salinity/ heat	LDH protection	[17,134]
*Ipomoea pescaprae*	DHN	Cold/drought/salinity	Antioxidant enzyme system upregulation	[135]
*Maize*	DHN11	Cold/drought/salinity	antioxidant enzymes activity protection	[136]
	DHN13	Cold	LDH protection	[16]
*Medicago truncatula*	DHN1	Cold/salinity	Membrane stabilization	[137]
	CAS31	Drought	Stomatal density and root water loss reduction	[120,138]
*Rice*	DHN1	drought/salinity	Enhancement of ROS scavenging capacity	[125]
*Saussurea involucrata*	DHN	Cold/drought	Inhibition of cell membrane damage, chloroplast protection, enhancement of ROS scavenging capacity	[139]
*Wheat*	COR410	Cold/drought	Plasma membrane protection against freezing and dehydration stress	[140]
	DHN5	Cold/salinity	LDH and bglG protection, and regulation of proline metabolism and ROS scavenging system	[126,141]

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
