# Peer review of "Plant Dehydrins: Expression, Regulatory Networks, and Protective Roles in Plants Challenged by Abiotic Stress"

_ijms, 2021, doi:10.3390/ijms222312619_

Round 1

Reviewer 1 Report

I recommend the article entitled ,,Plant Dehydrins: Expression, Regulatory Networks, and Protective Roles in Plants Challenged by Abiotic Stress” for publication in the International Journal of Molecular Science. The article is well written and contains important information about dehydrins in plants. Many references were used, of which 57% were from the last five years and 76 % were from the last 10 years.

I only have minor comments. The chapter Conclusions and Perspectives should be without references. At the end of the paragraph, the citation source should be marked for example in the Introduction line 31.

In Reference – position 88, the year of the publication should be bold.

Author Response

Dear Professor,

  We sincerely thank you for your affirmation and encouragement of our article. As you mentioned, our review "Plant Dehydrin: Expression, Regulatory Network and Protection in Plants Challenged by Abiotic Stress" keeps up with the latest research progress and focuses on recent research hotspots. And we also give your many thanks for thoroughly examining our manuscript and providing very helpful comments on how to improve our manuscript. We have tried our best to revise the manuscript according to your kind suggestions. The detailed changes are given below.

  1. We cleaned the references in the Conclusions and Perspectives and checked the use of citations in the whole manuscript and marked the citation at the suitable place as the suggestion.
  2. Based on your question on the format of the reference, we bolded the year of publication in 100th reference. And meanwhile we have checked and modified each reference one by one.

  In addition, we have made a small change in the positions of the two corresponding authors. Because Prof. Wang is the first corresponding author, we hope that you will give us tolerance and understanding.

  We sincerely hope that this revised manuscript can meet your requirements.

  Once again, thank you for your good suggestions and comments.

  We look forward to hearing from you.

Reviewer 2 Report

Dear Authors

This is a review based on the physiological and biological role of plant dehydrin and its influence on gene expression.

The review is correctly organized and easily readable.

Figures are clear, but I suggest to add some structures showing the main hormones and/or organic structures involved in this metabolic pathway.

Considering the growth of interest in natural organic compounds from plants exerting a protective role , I suggest to improve the introduction with some other agents such as those from robisco, e.g. rbiscolin and its analogues. Take a look to the literature, e.g. Stefanucci et al. 2020 J functional foods, 73, 104154.

Author Response

Dear Professor,

  We sincerely thank you for generalization and affirmation of our article., and so many suggestions and comments, which are of great help to improve our manuscript. Now we have made the following revision.

  1. We have supplemented ABA-receptor-PP2C complex structure (Figure 2A) and the action model of the ABA core pathway (Figure 2B), which make it easy to comprehense how the ABA signalling regulate the expression of dehydrins.
  2. In introduction, we added one paragraph in which we summarized the protective roles of some antioxyditive enzymes and natural organic compounds,such as SOD, APX, active peptides (rubiscolin-6), vitamin E, vitamin C, polyphenols and anthocyanins.

  In addition, we have made a small change in the order of the two corresponding authors. Because Prof. Wang is the first corresponding author, we hope that you will give us tolerance and understanding.

  We are appreciated for your suggestions and hope that the correction can meet your requirements.

  Once again, thank you for your good suggestions and comments.

  We look forward to hearing from you.

Reviewer 3 Report

In this manuscript, the authors did review the plant dehydrins: expression, regulatory networks, and protective roles in plants challenged by abiotic stress. Revie is important since dehydrins are essential members of the late embryogenesis abundant (LEA) protein family and are characterized by lysine-rich K-segments. In addition, a wide range of hostile environmental conditions, including low temperature, drought, and high salinity, stimulate dehydrin expression. Dehydrins also play essential roles in seed maturation and plant stress tolerance. Hence, dehydrins might also protect plasma membranes and proteins and stabilize DNA conformations. In the present review, the authors discuss the regulatory networks of dehydrin gene expression, including abscisic acid (ABA), mitogen-activated protein (MAP) kinase cascade, and Ca2+ signaling pathways. Crosstalk among these molecules and pathways may form a complex, diverse regulatory network that regulates the same Dehydrin.

I found many drawbacks in this review:

  1. The manuscript lack mention of Dehydrin in the most important crop plant, rice, and others like cucumber; it should cite reference such as a. Over-expression of dehydrin gene, OsDhn1, improves drought and salt stress tolerance through scavenging of reactive oxygen species in rice (Oryza sativa). b. Genome-Wide Identification of the Dehydrin Genes in the Cucurbitaceae Species.
  2. There are many reviews about plant dehydrins in recent times, Such as a. Frontiers | Multifunctional Roles of Plant Dehydrins in Response to Environmental Stresses | Plant Science (frontiersin.org). b.Dehydrin in the past four decades: From chaperones to transcription co-regulators in regulating abiotic stress response - ScienceDirect. What is new in this review?
  1. The manuscript is written poorly and needs serious English editing; Such as at L33 wesre to were.
  2. Manuscript is plagiarized at L10-11, L37-38, L43-44, L54-55, L72-73, L104-106, L131-133, L148, L149, L193-194, L211, L222-223, L272-273, L313-314, L332-333, L337, L341-342, L348-349, L355-356, L371-372, L373-375, L426-427. Please clean it.
  3. I Like figure's quality.

Author Response

Dear Professor,

  Sincere thanks should be given to you for the constructive suggestions and comments. Our responses to your questions and suggestions are shown below.

  1. We introduced some researches in recent years on dehydrins in crop plants such as rice, maize, cucumber and melon. We add them in the section of classification of dehydrins, dehydrins protect enzymatic activity, and the function of dehydrins in different plants at table 1. Specially, we read the two articles you provided and cited them in references.
  2. Thank your recommendation of the reviews. We read these papers and we also learned a lot new knowledge about dehydrins. Liu et al. (2017) have summarized the different molecular mechanisms of dehydrins response to environmental stress in plants. Tiwari et al. (2021) provided a linked and comprehensive introduction of dehydrins gene family from its discovery to now. In these reviews authors detailedly described the functions and molecular mechanisms of dehydrins response to abiotic stress. However, as we know, the article focusing on upstream regulation mechanism of dehydrins has been rarely published. Our manuscript emphatically introduced the upstream signal pathways how to regulate the expression of dehydrins under abiotic stress.
  3. We feel very apologetic about our manuscript exist English editing problems and thank you so much for pointing out it. We immediately checked the grammar and spelling issues in the manuscript.
  4. We have modified the sentences similar to those in publication.

 In addition, we have made a small change in the order of the two corresponding authors. Because Prof. Wang is the first corresponding author, we hope that you will give us tolerance and understanding. 

  At last, we hope that this revised manuscript can meet your requirements.

Round 2

Reviewer 3 Report

I am happy with the author's reply and the manuscript have improved a lot and I think it can be accepted in its current form for publication.